# Local Anesthesia versus Conscious Sedation among Patients Undergoing Transcatheter Aortic Valve Implantation—A Propensity Score Analysis

**DOI:** 10.3390/jcm11113134

**Published:** 2022-05-31

**Authors:** Anat Berkovitch, Ariel Finkelstein, Israel M. Barbash, Ran Kornowski, Paul Fefer, Arie Steinvil, Hana Vaknin Assa, Haim Danenberg, Elad Maor, Victor Guetta, Amit Segev

**Affiliations:** 1Division of Cardiology, Leviev Heart and Vascular Center, Chaim Sheba Medical Center, Tel-Hashomer 5262000, Israel; anatberko@gmail.com (A.B.); israel.barbash@sheba.health.gov.il (I.M.B.); paul.fefer@sheba.health.gov.il (P.F.); elad.maor@sheba.health.gov.il (E.M.); victor.guetta@sheba.health.gov.il (V.G.); 2Sackler School of Medicine, Tel-Aviv University, Tel Aviv 69978, Israel; afinkel@tlvmc.gov.il (A.F.); ran.kornowski@gmail.com (R.K.); ariksteinvil@gmail.com (A.S.); hana100niki@gmail.com (H.V.A.); 3Division of Cardiology, Tel Aviv Medical Center, Tel Aviv 6423906, Israel; 4Division of Cardiology, Rabin Medical Center, Petach-Tikva 4941492, Israel; 5The Heart Institute, Hadassah Ein-Karem Medical Center, The Hebrew University, Jerusalem 91120, Israel; danen040@gmail.com

**Keywords:** aortic valve, transcatheter aortic valve implantation, valvular disease

## Abstract

Background: Conscious sedation (CS) has been used successfully to treat patients with severe aortic stenosis (AS) undergoing transcatheter aortic valve implantation (TAVI) and as such is considered the standard anesthesia method. The local anesthesia (LA) only approach may be feasible and safe thanks to improvements in operators’ experience. Objective: To evaluate differences between LA only versus CS approaches on short- and long-term outcomes among patients undergoing TAVI. Methods: We performed a propensity score analysis on 1096 patients undergoing TAVI for severe AS. Two hundred and seventy-four patients in the LA group were matched in a ratio of 1:3 with 822 patients in the CS group. The primary outcome was a 1-year mortality rate. Secondary outcomes included procedural and peri-procedural complication rates and in-hospital mortality. Results: Patients in the CS group had significantly higher rates of grade 2–3 acute kidney injury and were more likely to have had new left bundle branch block and high-degree atrioventricular block. Patients who underwent TAVI under CS had significantly higher in-hospital and 1-year mortality rates compared to LA (1.6% vs. 0.0% *p*-value = 0.036 and 8.5% vs. 3.3% *p*-value = 0.004, respectively). Kaplan–Meier’s survival analysis showed that the cumulative probability of 1-year mortality was significantly higher among subjects undergoing CS compared to patients LA (*p*-value log-rank = 0.024). Regression analysis indicated that patients undergoing CS were twice more likely to die of at 1-year when compared to patients under LA (HR 2.18, 95%CI 1.09–4.36, *p*-value = 0.028). Conclusions: As compared to CS, the LA-only approach is associated with lower rates of peri-procedural complications and 1-year mortality rates.

## 1. Introduction

There is a growing trend towards a minimalistic approach in patients undergoing transcatheter aortic valve implantation (TAVI). General anesthesia and conscious sedation (CS) have both been used successfully to treat these patients. [1]. Compared with general anesthesia, CS is associated with a shorter length of stay and lower in-hospital and 30-day mortality [2,3] resulting in the wide adoption of CS as the primary method of anesthesia for TAVI procedures. [4].

The LA-only approach is also feasible thanks to improvements in operators’ experience [5], and was previously found to be associated with a reduced length of stay [6]. We aimed to evaluate differences between LA-only versus CS approaches on short- and long-term outcomes among patients undergoing TAVI. 

## 2. Methods

The study population included patients who underwent TAVI at four highly experienced tertiary medical centers in Israel between January 2010 and December 2019. All subjects were referred to TAVI after a careful evaluation by each institutional heart team. Baseline and procedural parameters were recorded in a computerized database using REDCap electronic data-capture tools hosted at The Israeli Center for Cardiovascular Research. All subjects underwent a detailed echocardiography before and after the procedure.

### 2.1. Study Groups and Definitions

Subjects were divided into two groups based on the TAVI procedure anesthesia method, CS, and local anesthesia (LA). The anesthesia method was decided by each center and operator according to patient status and local expertise. 

The regional ethical review board at each site approved of the trial protocol, and the trial was conducted according to the principles of the Declaration of Helsinki. Institutional review board approval was obtained from all the participating centers and all patients provided signed informed consent to participate in the study.

History of ischemic heart disease, hypertension, diabetes mellitus, and history of stroke were extracted from patients’ electronic medical history files based on known diagnoses or concurrent diabetic or blood-pressure lowering medications. Their renal function was evaluated using the Modification of Diet in Renal Disease equation. Peri-procedural outcomes and complications were recorded according to the Valve Academic Research Consortium-2 [7].

The primary outcome of the current study was a 1-year mortality rate. Secondary outcomes included procedural and peri-procedural complication rates and in hospital mortality. Mortality rates were ascertained with the Israeli ministry of interior mortality database.

### 2.2. Statistical Analysis

A propensity matched analysis was then performed using the nearest neighbor method by comparing patients in the LA group to patients in the CS group. Parameters that were found to be significant in the univariate model or that are known to be significant in the survival of patients undergoing TAVI were incorporated into the matching model. The matching included the following variables: age, gender, ischemic heart disease, prior coronary artery bypass graft surgery, diabetes mellitus, hypertension, previous stroke, Euroscore 2 and chronic kidney disease. Patients who underwent TAVI under LA were matched to patients in the CS group, using individual propensity scores, in a 1:3 ratio.

Continuous data were compared with Student’s t-test and a one-way ANOVA. Categorical data were compared with the chi-square test or the Fisher exact test. 

The probability of 1-year mortality by the pre-specified TAVI groups was estimated and graphically displayed according to the Kaplan–Meier method, with a comparison of cumulative events across strata by the log-rank test. Cox proportional hazard regression modeling was used to evaluate hazard ratios for 1-year mortality.

Statistical significance was accepted for a two-sided *p* < 0.05. The statistical analyses were performed with IBM SPSS version 25.0 (Chicago, IL, USA) and with Rstudio version 1.2.1335.

## 3. Results

A total of 1096 patients were included in the current study. A total of 204 patients in the LA group were matched in a ratio of 1:3 with 822 patients in the CS group. The mean age of the study population was 81 ± 7, of whom 58% were female. Clinical characteristics were generally well balanced between the groups (Table 1).

Patients in the CS group had a statistically significantly higher ejection fraction (56% vs. 52%, *p*-value = 0.037) and higher systolic pulmonary artery pressure (39 vs. 34 mmHg, *p* < 0.001). Other echocardiography parameters were similar between the two groups including the aortic valve area (0.7 cm^2^ for both), mean pressure gradient (44 mmHg), peak pressure gradient (70 mmHg) and left atrial area (25 cm^2^) (Table 2).

### Procedural and Peri-Procedural Outcome

Most patients underwent TAVI using a femoral access (99%, Table 3). Self-expandable valves were used in the majority of patients in both groups (58% and 60%). No significant differences were found in valve size between the two groups (Table 3). Patients in the LA group were more likely to have had balloon pre-dilatation (81% vs. 61%, *p*-value < 0.001), while patients in the CS group more often presented with post-dilatation (72% vs. 78%, *p*-value = 0.027). Both groups had similar high procedural success rates.

The peri-procedural complication rates are displayed in Figure 1. Of note, patients in the CS group had significantly higher rates of grade 2–3 acute kidney injury (2.5% vs. 0.0%, *p*-value = 0.004), higher rates of at least moderate paravalvular leak at the end of the procedure (2.2% vs. 0%, *p*-value = 0.025) and were more likely to have had new left bundle branch block and high-degree atrioventricular block (25.2 vs. 16.8%, and 7.8% vs. 4.0%, *p*-value < 0.001 for both, respectively). No other differences were identified between the groups regarding procedural or peri-procedural complications (Figure 1). 

Patients who underwent TAVI under CS had significantly higher in-hospital and 1-year mortality rates compared to the LA group (1.6% vs. 0.0% *p*-value = 0.036 and 8.5% vs. 3.3% *p*-value = 0.004, respectively). Kaplan–Meier’s survival analysis showed that the cumulative probability of 1-year mortality was significantly higher among subjects undergoing TAVI under CS compared to LA (*p*-value log-rank < 0.001, Figure 2). The Cox regression analysis indicated that patients undergoing CS had 2-fold higher 1-year mortality rate as compared to patients under LA (HR 2.18, 95%CI 1.09–4.36, *p*-value = 0.028) (Table 4). The length of stay was similar between the two groups (5.0 days vs. 4.9 days, *p*-value = 0.95). 

## 4. Discussion

The current study conducted on a large cohort demonstrated that TAVI under LA is associated with a reduced peri-procedural complication rate and lower long-term mortality as compared to CS. 

A small study by Piayda et al. [6] evaluated 215 patients undergoing TAVI and found that LA only was associated with a significantly shorter length of stay in the intensive care unit compared to patients with additional sedation or general anesthesia. In this study, no other major differences were found between the LA only group and CS. This study has many limitations; however, it was unique and innovative, since it was the first to present the option of the LA-only approach during this stage.

### 4.1. Conscious Sedation vs. General Anesthesia

The differences in patient outcomes were evaluated numerous times [8]. The advantages of CS when compared to GA are indisputable and include a shorter length of stay [8,9], lower rates of in-hospital mortality rates [2], less bleeding and vascular events, and shorter intensive care unit hospitalization [10]. Conflicting results were found regarding mortality rates at 30-day and 1-year mortality. While some studies [1,3] found no significant difference between the two approaches, others [9] found CS to be superior to GA, with significantly lower rates of mortality. Furthermore, healthcare cost was significantly lower per patients in the CS group compared to GA [8]. While many papers [1,2,11,12,13,14] previously demonstrated CS to be superior to general anesthesia, to the best of our knowledge, this is the first study to have shown the LA only approach to be superior to CS with respect to a hard endpoint such as mortality, and with softer endpoint, which are associated with long term poor outcomes such as acute kidney injury, at least a moderate paravalvular leak and new left bundle branch block.

We speculate that hemodynamic changes caused by Midazolam and the blood-pressure lowering effect might be associated with higher rates of conduction disorders and acute renal failure among the CS group. Another theory is that the Midazolm vagolytic effect causes tachycardia that reveals conduction disorders in octogenarians’ hearts.

### 4.2. Local Anesthesia in Non-Cardiology Procedures

Other disciplines have found LA to be superior to CS. Among patients with acute ischemic stroke undergoing intra-arterial treatment, CS was associated with poor functional outcomes and increased mortality rates compared to LA [15]. It is speculated that changes in mean arterial blood pressure using sedation or general anesthesia affect the clinical outcome among such patients [16,17]. Gastroenterologists encounter sedation complications when performing endoscopies in elderly patients. Sedation during colonoscopy among such patients induces statistically significant decrease in arterial oxygen saturation and increases the risk of hypotension [18,19]. In order to improve the value of the procedure and facilitate access to this life-saving intervention within the budgetary constraints, there is an increasing need to reduce the post-procedural length of stay. Reducing LOS not only has economic benefits, but it serves as a way to reduce post-procedural adverse events among the elderly population such as infections, mal-nutrition, depression and debilitation [20]. The LA-only approach can help achieve this goal.

## 5. Limitations

The present study has several limitations. First, this is a historical prospective, non-randomized, non-blinded observational study; therefore, it is subjected to limitations inherent in its design. However, a propensity-score matching analysis was used in order to overcome some of the design pitfalls. Second, the anesthesia method was decided by the senior physician performing the procedure, and each physician’s experience and expertise could influence both the method used and the success of the procedure. However, the strategy was time dependent rather than patient dependent. Third, although the best available propensity score matching was used, other factors such as frailty could not be measured and calculated in the analysis.

## 6. Conclusions

In patients with severe aortic stenosis undergoing TAVI, the LA-only approach is associated with lower rates of peri-procedural complications and 1-year mortality rates as compared to CS.

## Figures and Tables

**Figure 1 jcm-11-03134-f001:**
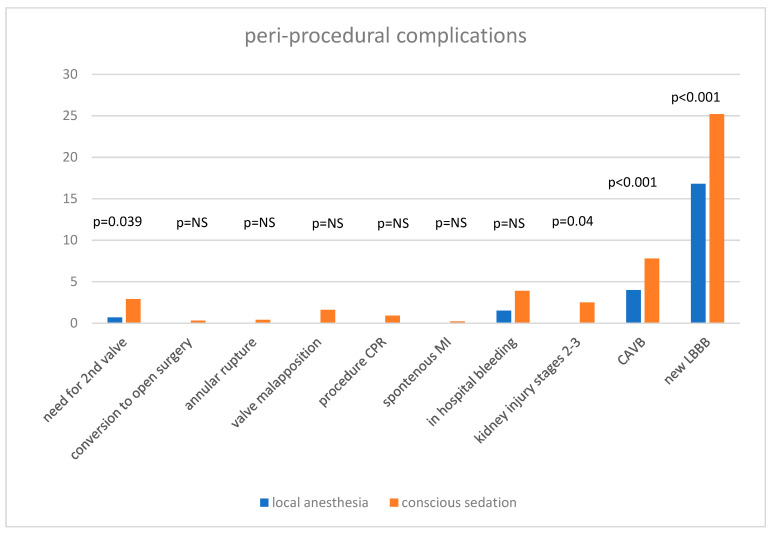
Peri-procedural complications.

**Figure 2 jcm-11-03134-f002:**
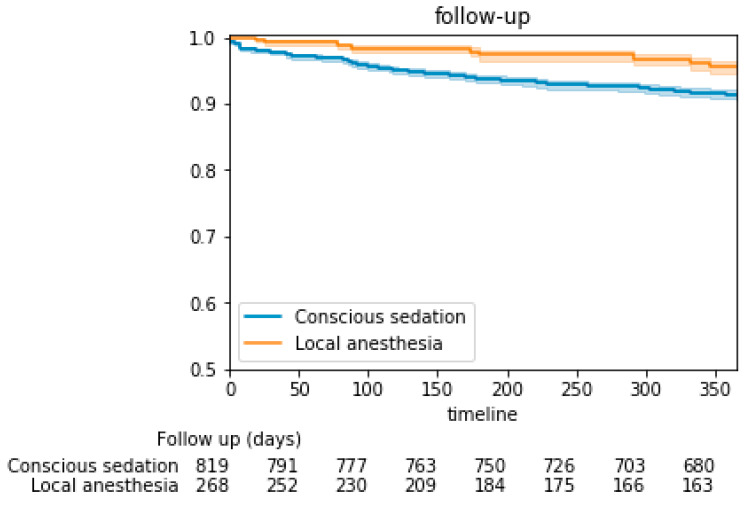
-Kaplan–Meier’s survival analysis according to the pre-specified study groups.The figure shows the cumulative probability of 1-year mortality according to the pre-specified study groups (*p*-value log-rank = 0.024).

**Table 1 jcm-11-03134-t001:** Patients baseline characteristics.

	Entire Cohort			Propensity-Score Matching Cohort		
	Local Sedation (N = 425)	Conscious Sedation (N = 2827)	*p*-Value	Local Sedation (N = 274)	Conscious Sedation (N = 822)	*p*-Value
Age, mean ± SD	81 ± 8	82 ± 7	<0.001	81 ± 7	81 ± 7	NS
Male, N (%)	220 (52)	1349 (48)	0.09	114 (42)	344 (42)	NS
Coronary artery disease, N (%)	181 (43)	1293 (47)	NS	174 (63)	533 (65)	NS
Prior myocardial infarction N (%)	30 (11)	436 (17)	0.004	30 (11)	431 (17)	NS
s/*p* CABG N (%)	75 (26)	474 (19)	0.002	70 (25)	206 (25)	NS
Past CVA/TIA, N (%)	54 (13)	401 (14)	NS	16 (13)	130 (16)	NS
Diabetes mellitus, N (%)	167 (41)	1077 (39)	NS	122 (44)	356 (43)	NS
Hypertension, N (%)	354 (87)	2353 (85)	NS	250 (91)	756 (92)	NS
Chronic kidney disease, N (%)	214 (50)	1091 (39)	<0.001	135 (49)	407 (49)	NS
Euroscore2, mean ± SD	4.2 ± 4.1	4.8 ± 9	0.024	4.7 ± 4.5	4.6 ± 4.5	NS
STS score, mean ± SD	4.1 ± 2.9	4.5 ± 3.4	0.01	4.4 ± 3.1	4.0 ± 2.5	NS
Body mass index, mean ± SD	27.8 ± 4.7	27.7 ± 5.0	NS	27.8 ± 4.6	28.1 ± 8.5	NS

Abbreviations: SD–standard deviation; CABG–coronary artery bypass graft; CVA–cerebrovascular accident; TIA–transient ischemia attack; NS–non-significant.

**Table 2 jcm-11-03134-t002:** Baseline echocardiography.

	Local Sedation (N = 274)	Conscious Sedation (N = 822)	*p*-Value
Simpson EF (%)	52 ± 11	56 ± 11	0.037
LA Area (cm^2^)	25.1 ± 6.7	24.8 ± 6	NS
Aortic valve Peak Pressure(mmHg)	70 ± 24	71 ± 24	NS
Aortic valve Mean Pressure(mmHg)	44 ± 16	44 ± 16	NS
Aortic valve area (cm)	0.7 ± 0.2	0.7 ± 0.2	NS
Systolic pulmonary artery pressure (mmHg)	34 ± 21	39 ± 17	0.001

Abbreviations: EF–ejection fraction; LA–left atrium; NS–non-significant.

**Table 3 jcm-11-03134-t003:** Procedural data.

	Local Sedation (N = 274)	Conscious Sedation (N = 822)	*p*-Value
Vascular access (Femoral artery)	272 (99%)	813 (99%)	NS
Valve type			
Self-expandable	130 (58%)	479 (60%)	0.07
Balloon-expandable	94 (42%)	306 (38%)	
Mechanically expandable	0 (0%)	16 (2%)	
Valve size			
23 mm	32 (12%)	123 (15%)	NS
26 mm	100 (36%)	300 (36%)	
29 mm	89 (32%)	32 (39%)	
34 mm	3 (1%)	28 (3%)	
Balloon pre-dilatation	218 (81%)	489 (61%)	<0.001
Balloon post-dilatation	193 (72%)	629 (78%)	0.027
Device success	260 (95%)	749 (96%)	NS

NS–non-significant.

**Table 4 jcm-11-03134-t004:** Cox regression analysis for 1-year mortality.

	Hazard Ratio	95% Confidence Interval	*p*-Value
Conscious sedation vs. local anesthesia	2.18	1.09–4.36	0.028

## Data Availability

The data presented in this study are available on request from the corresponding author.

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
