# Peer review of "Local Anesthesia versus Conscious Sedation among Patients Undergoing Transcatheter Aortic Valve Implantation—A Propensity Score Analysis"

_jcm, 2022, doi:10.3390/jcm11113134_

Round 1

Reviewer 1 Report

General anesthesia and conscious sedation in combination with local anesthesia have both been successfully used for TAVR, depending on the center preference. Most centers nowadays perform the implantation under fluoroscopy guidance and without transesophageal ultrasound. Thus a general anesthesia, which is more stress- and harmful, became unnecessary in most of the cases.

In their propensity score analysis the authors aimed to evaluate differences between local anesthesia and conscious sedation in Patients undergoing TAVR. How conscious sedation, without orotrachel intubation, can influence the outcome of TAVR is an interesting and important question and the authors must be congratulated to this approach. 

However, while reading the draft, I noticed a few points that I would ask the authors to revise:

1.) Regarding the Method, I would like to ask the authors about the initial strategy to perform Local anesthesia or conscious sedation. It could be speculated that this is a big confounder in the study as maybe patients undergoing conscious sedation had special reasons for this. Those could influence the whole study outcome.

2.) How the authors explain the fact, that patients in the CS group had significantly higher rates of LBB and AV- Block. One might assume that this is more an anatomical problem, which should be independent of the anesthetic method? Post-dilatation is a known risk factor for annulus rupture and arrhythmias. In the CS group was a higher number of post-Dilatations. May this explain the different rates in arrhythmias?

3.)I would like to ask the authors to discuss their findings point by point and more detailed. How do you explain the results. E.g. in the CS were significantly more patients with preoperative chronic kidney disease. How was the GFR? How are the values of creatinine and urea? This may explain the higher postoperative rates of acute kidney injury in the CS group.

4.)Could the authors identify other risk factors regarding primary and secondary outcome (except for the anesthetic method)?

Author Response

General anesthesia and conscious sedation in combination with local anesthesia have both been successfully used for TAVR, depending on the center preference. Most centers nowadays perform the implantation under fluoroscopy guidance and without transesophageal ultrasound. Thus, general anesthesia, which is more stressful- and harmful, became unnecessary in most of the cases.

In their propensity score analysis, the authors aimed to evaluate differences between local anesthesia and conscious sedation in patients undergoing TAVR. How conscious sedation, without orotrachel intubation, can influence the outcome of TAVR is an interesting and important question and the authors must be congratulated to this approach. 

However, while reading the draft, I noticed a few points that I would ask the authors to revise:

1.) Regarding the Method, I would like to ask the authors about the initial strategy to perform Local anesthesia or conscious sedation. It could be speculated that this is a big confounder in the study as maybe patients undergoing conscious sedation had special reasons for this. Those could influence the whole study outcome.

We thank the reviewer for this important comment. The initial strategy to perform local anesthesia or conscious sedation can indeed be a co-founder. However, the strategy was time-dependent rather than patient-dependent, meaning that a center decided to perform all TAVI patients using local anesthesia-only approach at a given time. This information was added to the revised manuscript.

This information was added to the limitations section in the revised manuscript, page 6, line 190.

2.) How the authors explain the fact, that patients in the CS group had significantly higher rates of LBB and AV- Block. One might assume that this is more an anatomical problem, which should be independent of the anesthetic method? Post-dilatation is a known risk factor for annulus rupture and arrhythmias. In the CS group was a higher number of post-Dilatations. May this explain the different rates in arrhythmias?

We thank the reviewer for enlightening our eyes on this issue. We speculate that hemodynamic changes caused by Midazolam and the blood pressure-lowering effect might be associated with this finding. Another theory is that Midazolam vagolytic effect causes tachycardia that reveals conduction disorders in octogenarians’ hearts. Post dilatation was significantly higher among the LA-only approach 28% vs. 21%, p-value=0.023).

This information was added to the discussion section in the revised manuscript, page 6, lines 164-168.

3.) I would like to ask the authors to discuss their findings point by point and more detailed. How do you explain the results. E.g. in the CS were significantly more patients with preoperative chronic kidney disease. How was the GFR? How are the values of creatinine and urea? This may explain the higher postoperative rates of acute kidney injury in the CS group.

As suggested in the previous comment, we speculate that renal failure is related to hemodynamic changes. CKD was one of the many PSM factors and as such rate of renal impairment at baseline was similar between the two groups (49.5% vs. 49.3%, p-value=0.094). Creatinine levels and GFR were also similar 0.83 mg/dl vs 0.76 mg/dl, p-value 0.56 and 23.7 ml/min/1.73m2 vs. 22.7 ml/min/1.73m2, p-value=0.04 (in favor of CS only), respectively.

This information was added to the discussion section in the revised manuscript, page 6, lines 164-168.

4.) Could the authors identify other risk factors regarding primary and secondary outcome (except for the anesthetic method)?

All known risk factors were matched using propensity score matching analysis to overcome this matter.

Reviewer 2 Report

please check the pdf. file

Author Response

There is a growing trend towards a minimalistic approach in patients undergoing 37 transcatheter aortic valve implantation (TAVI). General anesthesia and conscious sedation (CS) have both been used successfully to treat patients with severe aortic stenosis (AS) undergoing TAVI [1]. CS is associated with shorter length of stay and lower in-hospital and 30-day mortality in comparison to TAVI under general anesthesia[2,3]. As such, in most centers, CS with local anesthesia (LA) is considered the standard anesthesia method for TAVI procedures [4]. Check characters.

The instruction section was revised accordingly, page 1, lines 38-44.

Comments the lower numbers of alternative access, compared to other experience

Since we ruled outpatients under deep sedation and general anesthesia, alternative access such as trans-apical or trans-axillary were excluded from the current study. The local anesthesia approach is reserved, almost exclusively, for patients undergoing TAVI in the trans-femoral approach.

It is speculated that changes in mean arterial blood pressure using sedation or general anesthesia affect clinical outcome among such patients [16,17]. However is still open the debate regarding negative or positive effects of general anesthesia

We could not agree more, and many more studies would have to take place before any conclusion. However, we hope this study is the first to give rise to others to come.